# Two Novel *FAM20C* Variants in a Family with Raine Syndrome

**DOI:** 10.3390/genes11020222

**Published:** 2020-02-20

**Authors:** Araceli Hernández-Zavala, Fernando Cortés-Camacho, Icela Palma-Lara, Ricardo Godínez-Aguilar, Ana María Espinosa, Javier Pérez-Durán, Patricia Villanueva-Ocampo, Carlos Ugarte-Briones, Carlos Alberto Serrano-Bello, Paula Jesús Sánchez-Santiago, José Bonilla-Delgado, Marco Antonio Yáñez-López, Georgina Victoria-Acosta, Adolfo López-Ornelas, Patricia García Alonso-Themann, José Moreno, Carmen Palacios-Reyes

**Affiliations:** 1Laboratory of Cellular and Molecular Morphology, Section of Postgraduate Studies and Research, Escuela Superior de Medicina, Instituto Politécnico Nacional, Salvador Díaz Mirón esq. Plan de San Luis S/N, Miguel Hidalgo, Casco de Santo Tomas, Mexico City 11340, Mexico; araheza17@gmail.com (A.H.-Z.); feercortes@gmail.com (F.C.-C.); icelitpl@yahoo.com (I.P.-L.); 2Direction and Division of Research, Hospital Juárez de México, Av. Instituto Politécnico Nacional 5160, Magdalena de las Salinas, Gustavo A. Madero, Mexico City 07760, Mexico; raa_rga@yahoo.com.mx (R.G.-A.); jbonilla@cinvestav.mx (J.B.-D.); giviac@gmail.com (G.V.-A.); adolfolopezmd@gmail.com (A.L.-O.); jmoreno49@gmail.com (J.M.); 3Service of Clinical Pharmacology, Hospital General de México, Dr. Balmis 148, Doctores, Cuauhtémoc, Mexico City 06720, Mexico; anaesga@hotmail.com; 4National Institute of Perinatology, Calle Montes Urales 800, Lomas - Virreyes, Lomas de Chapultepec IV Section, Miguel Hidalgo, Mexico City 11000, Mexico; djavier40@gmail.com (J.P.-D.); pgalonsot@yahoo.com (P.G.A.-T.); 5Deparment of Ginecology, Hospital Juárez de México, Av. Instituto Politécnico Nacional 5160, Magdalena de las Salinas, Gustavo A. Madero, Mexico City 07760, Mexico; vopaty@hotmail.com; 6Department of Pathology, Hospital Juárez de México, Av. Instituto Politécnico Nacional 5160, Magdalena de las Salinas, Gustavo A. Madero, Mexico City 07760, Mexico; alcuboster@gmail.com (C.U.-B.); crls.serrbe@gmail.com (C.A.S.-B.); drapjss1@gmail.com (P.J.S.-S.); 7Department of Radiology & Imagenology, Hospital Juárez de México, Av. Instituto Politécnico Nacional 5160, Magdalena de las Salinas, Gustavo A. Madero, Mexico City 07760, Mexico; marcomarx@gmail.com

**Keywords:** lethal Raine syndrome, *FAM20C*, new variants, histopathology

## Abstract

Two siblings from a Mexican family who carried lethal Raine syndrome are presented. A newborn term male (case 1) and his 21 gestational week brother (case 2), with a similar osteosclerotic pattern: generalized osteosclerosis, which is more evident in facial bones and cranial base. Prenatal findings at 21 weeks and histopathological features for case 2 are described. A novel combination of biallelic *FAM20C* pathogenic variants were detected, a maternal cytosine duplication at position 456 and a paternal deletion of a cytosine in position 474 in exon 1, which change the reading frame with a premature termination at codon 207 and 185 respectively. These changes are in concordance with a negative detection of the protein in liver and kidney as shown in case 2. Necropsy showed absence of pancreatic Langerhans Islets, which are reported here for the first time. Corpus callosum absence is added to the few reported cases of brain defects in Raine syndrome. This report shows two new *FAM20C* variants not described previously, and negative protein detection in the liver and the kidney. We highlight that lethal Raine syndrome is well defined as early as 21 weeks, including mineralization defects and craniofacial features. Pancreas and brain defects found here in FAM20C deficiency extend the functional spectrum of this protein to previously unknown organs.

## 1. Introduction

Raine Syndrome (RS) (OMIM # 259775) is an autosomal recessive lethal disease, first described in 1989 [1], with a prevalence of <1 in 1,000,000 [2]. RS is characterized by generalized osteosclerosis with periosteal bone formation, characteristic facial dysmorphisms, and intracerebral calcifications. Most of those affected by RS die within the first days or weeks of life due to pulmonary hypoplasia [3,4,5]. Although most cases are detected at birth, facial alterations such as flat facial profile, hypoplastic nose and prominent eyes have been described prenatally, in addition to cerebral alterations such as large choroid plexuses, echogenic appearance of the brain, ventricular blurry side walls, and intracerebral calcifications [6,7,8,9,10].

Pathogenic variants in *FAM20C*, were identified as the cause of RS in 2007 [11]. *FAM20C* gene, located in 7p22.3, belongs to the family with sequence similarity 20 (FAM20), which is constituted by three members: *FAM20A*, *FAM20B,* and *FAM20C* [12] that encode protein kinases acting on diverse substrates that play important roles in biomineralization [13]. The FAM20C protein is a casein kinase expressed in the Golgi apparatus that phosphorylates serine residues at S-X-E/pS motifs of proteins found in serum, plasma and cerebrospinal fluid. Secreted proteins phosphorylated by FAM20C are estimated to be over 100 genuine substrates [12,14]. They participate in processes that include wound healing, lipid homeostasis, endopeptidase inhibitory activity, adhesion and cell migration, and they also appear to be involved in cancer [14,15,16,17]. FAM20C role in the biomineralization process occurs through phosphorylation of members of the secretory calcium-binding phosphoproteins (SCPP) family, which include the small integrin-binding ligand N-linked glycoprotein (SIBLING proteins), such as dentin matrix proteins (DMP1), bone sialoprotein (BSP), osteopontin (OPN), matrix extracellular phosphoglycoprotein (MEPE), and dentine sialophosphoprotein (DSPP) [15,18,19,20]. Moreover, FAM20C regulates fibroblast growth factor 23 (FGF23) secreted by osteoblasts and osteocytes, which plays a major role in the renal metabolism of phosphate, in the reabsorption of phosphate and the catabolism of 1,25-dihydroxyvitamin D3 (1,25(OH)2D3) [21,22].

Prior to 2009, only lethal cases of RS had been described with the typical RS phenotype. The first non-lethal (NLRS) cases were described in 2009 [23], with new cases found in adolescents and adults with a wide variable expression [23,24,25,26,27,28,29,30,31,32,33] The major NLRS features are: hypoplastic nose, amelogenesis imperfecta, hearing defects, ectopic calcifications, osteonecrosis, and intellectual disability [3,4,5,10,23,24,25,26,27,28,29,30,31,32,33]. Most cases of lethal RS (LRS) are detected at birth, which present major features including facial alterations such as flat facial profile, hypoplastic nose, and prominent eyes, in addition to mineralization defects like osteosclerotic bone defects and vascular and brain calcification, as well as respiratory defects like choanal atresia/stenosis and lung hypoplasia [6,7,8,9,10,11,34,35,36,37]. In addition, features including polyhydramnios, facial profile similar to Binder anomaly and cerebral alterations such as large choroid plexuses, echogenic appearance of the brain, ventricular blurry side walls and intracerebral calcifications have been described prenatally [6,7,8,9,10]. Histopathological findings derived from RS necropsies have revealed the presence of calcifications such as calcospherites within the prevascular and perivascular brain neuropil, plus pulmonary hypoplasia, and abnormal bone mineralization including trabecular bone covered by abundant osteoid, plentiful osteoblasts, bone resorption and remodeling, and osteomalacia [7,8,35,36].

The purpose of the present work is to describe two new *FAM20C* gene pathogenic variants in a Mexican family with two siblings with lethal RS: a term newborn (case 1) and a 21 gestational week (GW) fetus (case 2). We also report, prenatal findings for case 2 and new pathological findings in the pancreas and FAM20C immunohistochemical staining in liver and kidney in case 2. We discuss and list *FAM20C* pathogenic variants in lethal Raine syndrome and its effects through different targets.

## 2. Materials and Methods

### 2.1. Cases 1 and 2

Two siblings from a Mexican family of apparently healthy parents, both of them 29 years old at the time of birth of the first case. They had no history of consanguinity or inbreeding (Figure 1A). The mother´s first pregnancy (G1) (from a different father), is now a nine-year old healthy daughter. Her second pregnancy (G2) corresponds to case 1, a term newborn male with clinical data compatible with RS who died on his third day of life. Case 2 corresponds to her next pregnancy (G3), detected prenatally by ultrasound, showing features compatible with RS at 21 weeks of gestation (GW). Both brothers’ weight, length, and head circumference were normal according to percentiles for their gestational ages. Informed consent was obtained from both parents and this work was approved by the ethics committee of the Hospital Juárez de México on July 13th, 2018 (Ethical code: HJM 0445/18-I). All the people involved in this study gave their informed consent in writing, with alignment to the Declaration of Helsinki 1975. For case 1, only clinical-radiological aspects are reported. For case 2, necropsy data are also reported with immunohistochemistry on sections of liver and pancreas. A sequence analysis of *FAM20C* exons was carried out in both parents.

### 2.2. Genetic Analysis

In order to identify *FAM20C* variants (NM_020223.3), we performed sequence analysis by polymerase chain reaction (PCR) and bidirectional Sanger sequencing using BigDye Terminator v3.1 kit (Applied Biosystems, Foster city, CA, USA) of exons 1–10 on peripheral blood DNA from both parents. Primers used were those reported by Acevedo et al. [31]. Analyses were done on the parents. Detected variants were searched in public databases (i.e., gnomAD, ExAC, 1000 Genomes, and ESP) [38,39,40,41].

### 2.3. Immuno-Histochemical Staining for FAM20C

Formalin fixed paraffin embedded tissue sections (5 µm thickness) from kidney and liver derived from case 2 and necropsy tissues of an unrelated 31 GW newborn without RS (control) (deceased due to non-syndromic congenital heart malformation), were used for immunohistochemical staining. We used liver and kidney sections because FAM20C expression is higher in these tissues (at least in adults), and because of his activity in FAM20C targets synthesized and secreted from hepatocytes, and its participation on regulating excretion of phosphate through binding to FGFRs/α-KL in kidney. Endogenous peroxidase activity was blocked by incubating the slides with 3% hydrogen peroxide in PBS for 10 min (Peroxide Blocking Reagent, catalogue: 927401 BioLegend, San Diego, CA, USA). Then, a non-specific background blocker was added and incubated for 10 min with a protein blocking reagent (Background Sniper, Catalogue: BS966 H, JJ, L, M, MM. Biocare Medical, Pacheco, CA, USA). Tissue sections were incubated with anti-FAM20C antibody diluted 1:50 (rabbit polyclonal, Concentration 100 µL at 0.72 mg/mL, Abcam, Cambridge, UK, catalog number ab154740). This antibody identifies the fragment corresponding to a region within amino acids 233-512 of Human FAM20C (UniProt ID: Q8IXL6). It was used by Cozza G and Knab VM (16, 71). For the negative control immunostaining, the primary antibody was replaced by PBS. Antigen-antibody complexes were detected by avidin-biotin-peroxidase with a Starr Trek Universal HRP Detection System KIT (Catalogue: STUHRP700 H, L10, Biocare Medical) following the supplier’s recommendations, and were counterstained with hematoxylin, and finally assembled with Entellan. Assays were performed in triplicate and photographed under a light microscope (Zeiss model 473028, Carl Zeiss, Oberkochen, Germany).

## 3. Results

### 3.1. Case Reports

Case 1 was a term newborn male from G2 uncomplicated pregnancy, without exposure to teratogens, who was delivered by cesarean section due to double loop nuchal chord. Craniofacial alterations and respiratory distress were readily detected at birth. The craniofacial phenotype was typical of RS (Figure 1B), showing turribrachycephaly, wide fontanelles, prominent frontal bone, flat facial profile, severe ocular proptosis, hypertelorism, depressed and low nasal bridge with hypoplastic nose, choanal stenosis, flaring nares, fish-like mouth, small pointed chin, rounded and low-set ears, and horizontal antitragus. Among extra-craniofacial features, he had a short, small, and slightly bell shape thorax; hands and feet presented brachydactyly, hypoplastic distal phalanges, and fingerpads. Radiologic imaging showed generalized osteosclerosis (skull, vertebras, thorax, long bones) and periosteal reaction (Figure 2A). A transfontanellar ultrasound study revealed brain calcifications (Figure 2B). The patient remained in the neonatal intensive care unit for 3 days with oxygen support, after which he developed seizures and cardio-respiratory arrest. RS diagnosis was done on the basis of osteosclerosis and craniofacial features, and parents were given genetic counseling with a 25% risk for future offspring.

Case 2 corresponds to G3. Pregnancy was assessed by structural ultrasonography since week 11. Facial dysmorphisms: ocular proptosis, hypertelorism, and mouth features were detected at 21 GW (Figure 1C), which lead to the decision to induce delivery. Brother 2 had also turricephaly, high and slightly bulged frontal, ocular proptosis, depressed nasal bridge, small nose, V palate, micrognathia, and rounded ears. Among non-craniofacial features, he had prominent fingerpads and undescended testicles. Babygram showed generalized osteosclerosis with a pattern in the base of the skull and facial bones, similar to case 1 (Figure 2C).

### 3.2. Pathological Features

Histopathological descriptions were reported for lethal RS (necropsy) and non-lethal (bone and gum tissues). Of the cases presented here, necropsy was carried out for case 2, who was described as a 22 GW male, weight 450 g, length 28.7 cm, head circumference 19 cm, thoracic perimeter 14.6 cm, abdominal perimeter 16 cm, and feet 4.2 cm. The brain weighed 71 g and measured 7 × 6 × 4cm, with lissencephaly and agenesis of corpus callosum (Figure 3A). Histological sections of the bone showed hyaline cartilage and immature trabecular bone. Brain sections showed disorganization of the cortical layers and multiple zones of microcalcifications (Figure 3A–C). In the pancreas, islets of Langerhans were absent (Figure 3D). No other tissue alterations were detected.

### 3.3. FAM20C Sequence Analysis

Sanger sequencing revealed two previously unreported variants in RS (reference sequence NM_020223), a duplication at position 456 (c.456dupC) in exon 1 and a deletion of a cytosine in position 704 also in exon 1 (c.474delC), shown in Figure 4. The first one alters the reading frame, resulting in a premature termination codon at position 207 (p.Gly153ArgfsTer56), which corresponds to the maternal sequence. The second is of paternal origin and also changes the reading frame due to a premature stop codon at codon 185 (p.Ser159ProfsTer28). The two cases would have inherited these variants and therefore they appear to be compound heterozygous. The fact that they are different variants, confirms the history of non-consanguinity. The variants found in both parents, have not been described in patients or families affected with RS, nor have they been reported as variants in ExAC, ESP, and 1000 Genomes databases [38,39,40,41]. GnomeAD found the paternal variant (p.Ser159ProfsTer28) in exome samples, which represents one allele from 134998 detected in Africans, Ashkenazi Jews, East Asian, European, South Asian, Latino, and other populations, with an allelic frequency of 0.000007408. This allele was detected in 23776 alleles from Latino population, with an allelic frequency of 0.00004206.

### 3.4. Immunohistochemical Detection of FAM20C

FAM20C indirect immunoreactivity was positive in the cytoplasm of tubular epithelial cells of the kidney in a 31 GW newborn control, whereas in our RS patient (case 2) there was no detection (Figure 5). In the liver, FAM20C indirect immunoreactivity was detected in control hepatocytes, but it was not detectable in RS patient. These results were done assuming a qualitative analysis.

## 4. Discussion

RS was initially described as a lethal syndrome within the first hours or weeks of life. Since the first report by Raine 30 years ago, 30 lethal cases have been described [1,6,7,8,9,11,34,35,36,37,42,43,44,45,46,47,48,49,50,51,52,53], besides our two cases. All individuals affected with lethal RS survive hours, days or weeks, with death mainly resulting from respiratory failure. In 2009 two unrelated individuals of 8 and 11 years were reported with non-lethal Raine syndrome, and to date, 22 cases have been described [3,4,5,10,23,24,25,26,27,28,29,30,31,32,33]. Thus, two types of RS are recognized: lethal and non-lethal RS. The cases reported herein have the classical features of LRS, mainly defined by mineralization defects and facial phenotype.

The craniofacial phenotype of case 1 had many of the classical features, such as wide fontanelles, prominent frontal bone, flat facial profile, severe ocular proptosis, hypertelorism, depressed and low nasal bridge with hypoplastic nose, and choanal stenosis. The second patient (case 2), had similar craniofacial features as his older brother but slightly less pronounced, probably due to a shorter gestational age (21 weeks). Among extracraniofacial features both had a short and narrow thorax, slightly bell shaped; brachydactyly, hypoplastic distal phalanges in hands, feet, and fingerpads, but also less pronounced in case 2. Cases 1 and 2 are radiologically compatible with RS: characterized by generalized osteosclerosis that was more prominent at the cranial base and facial bones, plus periostic reaction and hypoplastic distal phalanges in the hands. They also had carpal ossification defects (delayed bone age), not a common feature of RS. Babygram of case 2 also showed prominent osteosclerosis at the cranial base and facial bones. The earliest RS osteosclerosis reported was in a prenatal ultrasound at 23 GW, which in our case 2 it was present at 21 GW, making it the earliest known case of mineralization defects that could represent a striking and early LRS feature. Other described RS features like metaphyseal flaring, shortening and bowing of long bones, irregular contour of ribs, fractures, pseudofractures, flattened vertebral bodies, were not present in the cases described here. In summary, there were three major clinical and radiological components in our LRS cases: craniofacial phenotype, bone mineralization defects, and respiratory disturbances.

One of our patients had absence of the corpus callosum, an apparently uncommon feature, as neurological defects have been reported in two lethal cases, including encephalocele, cortical atrophy, cerebellar hypoplasia, and pachygyria [43,52], as well as two non-lethal cases including corpus callosum dysgenesis and cortical atrophy, apparent pituitary gland absence and posterior cerebellum hypoplasia [5,24].

Since its original description, RS was considered an autosomal recessive disorder and the responsible gene was detected in 2007 [11]. Table 1 summarizes *FAM20C* pathogenic variants for LRS reported to date (according to RefSeq NM_020223.3) [11,34,44,46,47,48,52,53].

In the present cases, sequencing analyses revealed that both parents were heterozygous carriers. Based on the effect of the two newly detected variants: duplication and deletion in exon 1, both yield a frameshift in the reading frame with a premature termination codon at codon 207 (maternal variant) and 185 (paternal variant), that predict 207 and 185 amino acid proteins [54]. On the basis of this, both variants can be clearly classified as pathogenic. The paternal variant has been detected in only one allele (allelic frequency of 0.000007408) at GnomAD, but it is not included in the ExAc, ESP and 1000 Genomes databases [38,39,40,41]. The maternal variant is not described in any of the main databases (GnomAD, ExAc, ESP and 1000 Genomes). We could not verify the presence of these variants in both cases due to DNA degradation of tissues in case 2 and unavailable tissue samples for case 1 as well as from the nine year old daughter. Nevertheless, as RS has an autosomic recessive inheritance, we can assume that both variants were inherited together in cases 1 and 2 (compound heterozygous). If this was the case, these two variants represent a novel combination of biallelic *FAM20C* variants in Raine osteosclerotic dysplasia and therefore, they are new variants associated to lethal RS, although one of them had been previously reported in only one heterozygous healthy individual.

The premature stop codon of both alleles is in concordance with a disease-causing loss of function variant. Moreover, although the gene product could undergo nonsense-mediated mRNA decay (NMD) [55,56], we could not corroborate it because RNA samples were not available. On the other hand, the FAM20C protein was not detectable by immunohistochemistry in kidney and liver tissues, which normally have a high expression according to The Human Protein Atlas database [56]. As the antibody used recognizes the region spanning aa 233–512, and both mutant proteins lack this region, these truncated proteins are not detectable, in addition to being non functional as predicted by the lack of the kinase domain located between aa 354–565 of wild type FAM20C. Therefore, the absence of any functional form of this protein could cause the lethal phenotype.

The fact that RS is a congenital syndrome, implies that FAM20C is important during embryonic development. The FAM20C protein importance in this period is supported by its detection in a 31GW control fetus. To our knowledge, FAM20C expression in these tissues was previously unknown. The phenotype of RS patients appears to be related to changes in the distribution of FAM20C and/or its kinase activity [57,58,59,60], which ranges from absent to varying degrees of impairment of S-x-E/pS motif phosphorylation of over 100 target proteins, besides the possible role of NMD. This would also explain the phenotypic spectra and the severity of lethal or non-lethal RS [15,59,60].

FAM20C participates in biomineralization through phosphorylation of different SCPP proteins, including the SIBLING and other proteins. SIBLING proteins include OPN, DMP1, BSP, extracellular MEPE, and DSPP. Other proteins include specific extracellular matrix and transport proteins, proteases and protease inhibitors, plus biologically active peptide hormones, which regulate calcium phosphate precipitation as hydroxyapatite as well as BMP4 (Bone morphogenetic protein 4), which has a role in osteoclast differentiation and maturation [19,20,61,62,63,64].

Some RS syndrome patients present elevation of FGF23, PTH and 1,25(OH)2D3, as well as hypophosphatemia with occasional hyperphosphaturia [21,26,27,28]. FAM20C deficiency causes mineralization abnormalities through FGF23, that regulates serum calcium and phosphate in the kidney though the NPT2a and NPT2c type II co-receptors of the proximal tubules, by decreasing phosphate reabsorption [64,65,66]. Moreover, it inhibits the synthesis of active 1,25(OH)2D3. Inactivation of FGF23 is mediated by its degradation in serum. Phosphorylation of FGF23 at Ser180 by FAM20C prevents O-glycosylation and, therefore, proteolytic cleavage by furin protease [22,66]. Hence, the absence of FAM20C increases plasma FGF23 activity. This leads to phosphate wasting, secondary hypophosphatemia, and low circulating 1,25(OH)2D3. This translates into hypophosphatemia and hypocalcemia or hypophosphatemic rickets in some RS patients [64,67,68].

Moreover, FGF23 probably induces parathyroid hormone (PTH) secretion by the parathyroid glands through its interaction with Klotho protein (KL), which is also induced by the reduction of serum calcium. PTH increases the reabsorption of calcium and phosphates and stimulates the production of 1,25(OH)2D3 by proximal tubular cells [66,68]. As a result of this, we suggest that all patients with unexplained bone sclerosis should have a metabolic bone screening.

In addition to bone tissue, FAM20C is expressed in teeth, causing severe dentin and enamel defects in RS [69,70]. Amelogenesis imperfecta (AI) and abnormal dentinogenesis are common features in NLRS [30,31], as most patients with lethal RS die within the first days of life, it is not possible to determine them in affected individuals as in cases 1 and 2, and to date only one lethal case that was two years old with teeth defects has been reported.

There could be additional, yet not identified FAM20C targets as suggested by the absence of Langerhans islets in case 2, which was not previously reported. Moreover, there are no data that indicate a role of FAM20C in pancreatic development. Finally, neurological involvement in NLRS, including one of our patients, suggests that FAM20C also plays a role in brain development and function. Although RS is presently defined by bone and mineralization disorders, the recent identification of several targets in different tissues give new clues to the understanding of its pathogenesis.

In summary, we present here two new pathogenic *FAM20C* variants in the parents of two siblings with LRS. These variants were apparently inherited as a biallelic combination in compound heterozygous, both of them causing a frameshift out of frame producing truncated proteins (185 and 207 aa). An anti-FAM20C could not detect the FAM20C protein in the liver and the kidney, suggesting that in the affected siblings only the truncated variants were present. It is also the first report of pancreas defects (lack of Langerhans islets) in RS. One of these patients had agenesis of the corpus callosum, which is known as a common feature. The phenotype of the present cases is highly representative of LRS, which include bone defects, such as osteosclerosis (highly striking in the skull base and facial bones), periostic reaction in long bones; the facial phenotype, plus defects in the brain and pancreas. FAM20C deficiency was evident early during embryonic development, as the phenotype was evident at 21GW. The addition of brain and pancreas development defects to RS will add to the understanding of the wide pathologic spectrum of RS.

## Figures and Tables

**Figure 1 genes-11-00222-f001:**
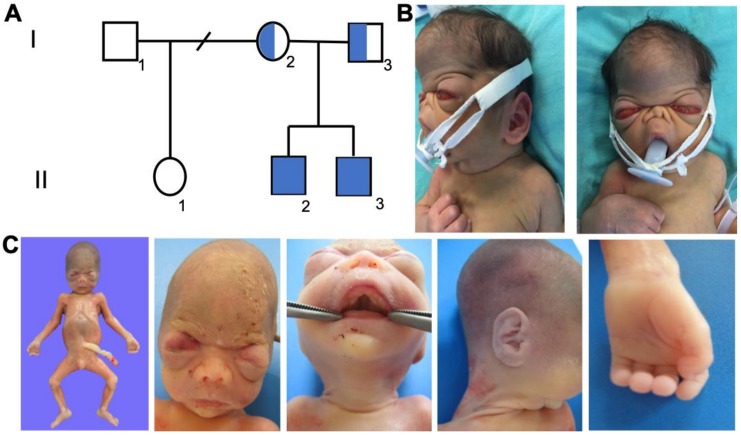
Family pedigree and phenotype of affected siblings (case 1 and 2). (**A**) Pedigree of the family showing reported cases (case 1 corresponds to individual II.2 and case 2 corresponds to individual II.3). (**B**) Clinical picture of the first proband (case 1). (**C**) Clinical pictures of the second patient (case 2).

**Figure 2 genes-11-00222-f002:**
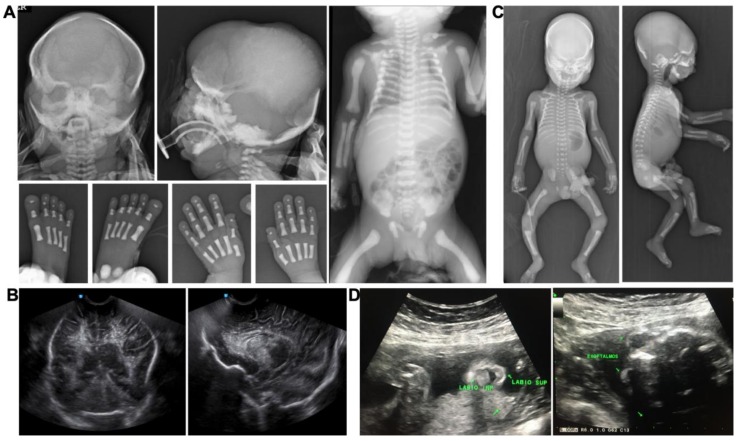
Image findings of cases 1 and 2. (**A**) X-ray of cranium, hands, and feet and babygram of the case 1, showing generalized osteosclerosis, more prominent in cranium base and facial bones, with hypomineralization of the other skull areas; periostic reaction in humerus, distal phalangeal hypoplasia of hands, carpal ossification defects (delayed bone age), and generalized osteosclerosis. (**B**) Transfontanellar ultrasound of case 1, showing brain calcifications, mostly periventricular. (**C**) Babygram of case 2, showing cranium with osteosclerosis, also more prominent in skull base and facial bones, cranial vault hypomineralization, and increased density of all the bones. (**D**) Obstetric structural USG at 21GW of case 2, showing exophthalmos and oral anomalies.

**Figure 3 genes-11-00222-f003:**
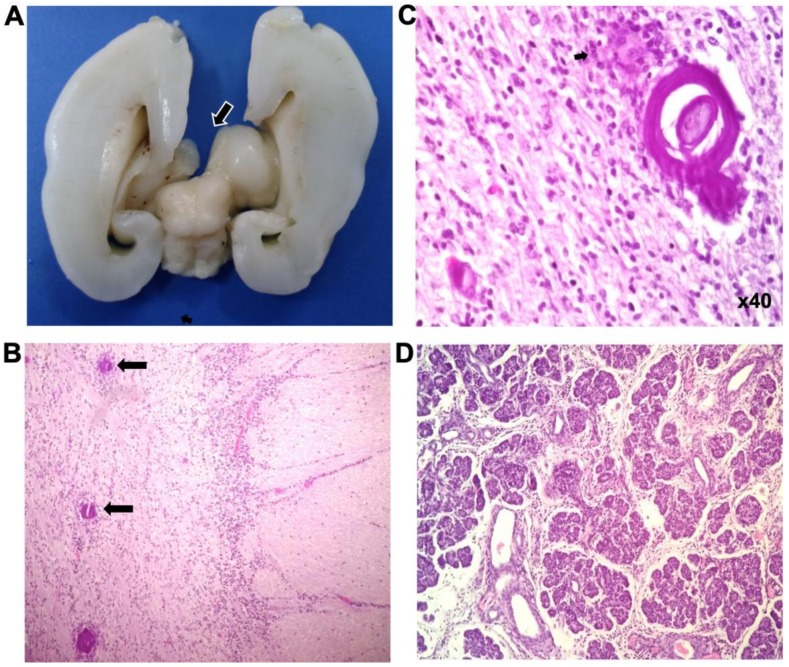
Anatomopathological findings of case 2. (**A**) Brain section showing corpus callosum agenesia. (**B**) Histological analysis of brain sections revealed brain parenchyma with disorganized cortex layers, and zones with laminated microcalcifications (arrows). (**C**) Neuropile with concentric calcospherites. (**D**) Pancreas with absence of Langerhans’ islets. (H&E stain).

**Figure 4 genes-11-00222-f004:**
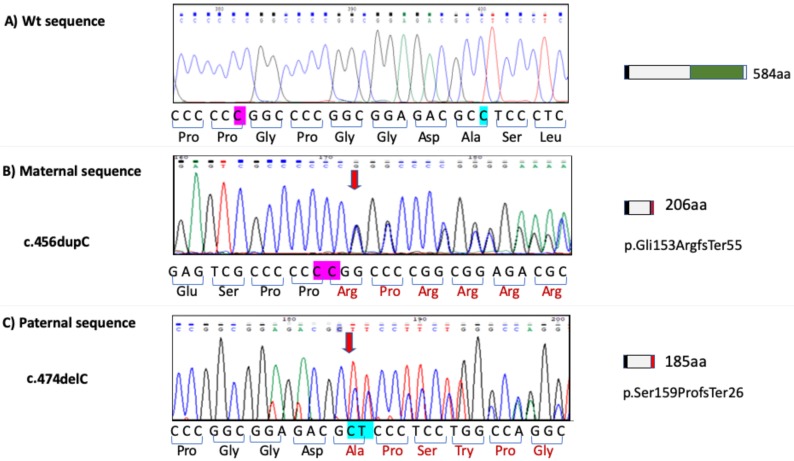
*FAM20C* gene sequence analysis from Family. (**A**) Electropherogram showing wild-type *FAM20C* sequence and localization sites of maternal and paternal variants. (**B**) Electropherogram showing identified maternal variant sequence c.456dup corresponding to p.Gly153Argfs*56 and effect on size protein. (**C**) Electropherogram showing paternal variant sequence c.474delC corresponding to p.Ser159Profs*28 and size protein effect.

**Figure 5 genes-11-00222-f005:**
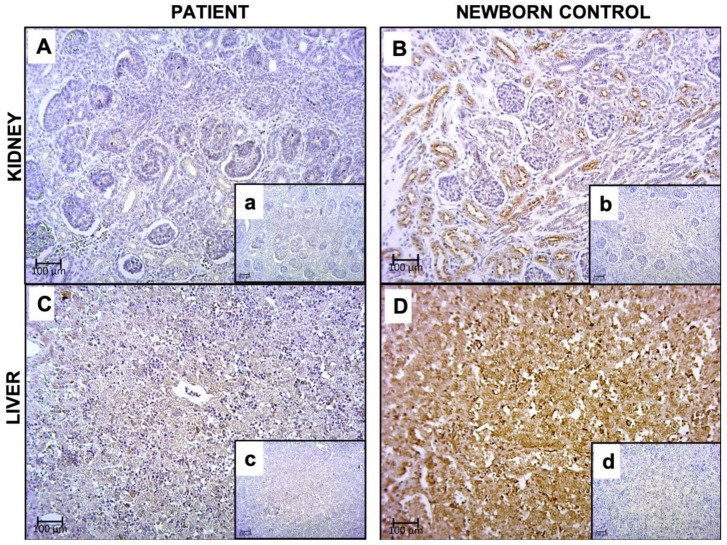
Representative images of FAM20C detection by immunohistochemistry (40×) in kidney and liver of patient with lethal Raine Syndrome (RS) (case 2) and necropsy tissue from a newborn without RS (gestational age 31 weeks). (**A**) Kidney from case 2, (a) negative control. (**B**) Kidney from newborn without RS (31GW), (b) negative control. (**C**) Liver from case 2, (c) negative control. (**D**) Liver from newborn without RS (31GW), (d) negative control.

**Table 1 genes-11-00222-t001:** *FAM20C* variants related to lethal Raine syndrome.

RC	Year	Reference	Exon /Localization	c.description	p.description	M	Sp	I/NS	GR	KD
1	2007	Simpson et al. [11]	7p22	45, XY psudic (7;7) (p22;p22)	-				X	X
2	2007/ 1991	Simpson et al. [11] /Kingston et al. [34]	E10	c.1645C > T	p.Arg549Trp	X				X
3	2007	Simpson et al. [11]	E6	c.1135G > A	p.Gly379Arg	X				X
4	2007/ 2003	Simpson et al. [11] Al-Gazali et al. [46]	I4–E5	c.957-3C > G	Splicing		X			-
5	2007/ 2003	Simpson et al. [11] Hülskamp et al. [47]	E6	c.1163T > G	p.Leu388Arg	X				X
6	2007	Simpson et al. [11]	E6/I7–E8	c.1136G > A/c.1364-2A > G	p.Gly379Glu/Splicing	X	X			X
7	2007	Simpson et al. [11]	E4–I4/I8–E9	c.956 + 5G > C/c.1446-1G > A	Splicing/Splicing		X/X			X/X
8	2010	Kochar et al. [48]	E10	c.1672C > T	p.Arg558Trp	X				X
9	2013	Ababneh et al. [50]	7p22 (48Kb)	46,XY.ar[hg19] 7p22.3 (36480−523731)×0	-				X	X
10	2015	Seidahmed et al. [52]	E6	c.1225C > T	p.Arg409Cys	X				X
11	2016	Whyte et al. [44]	E6	c.1094G > A	Gly365Asp	X				X
12	2016	Whyte et al. [44]	E6	c.1094G > A	Gly365Asp	X				X
13	2019	Hung et al. [53]	E5	c.1007T > G	p.Met336Arg	X				X

RC: Reported cases. M: Missense, Sp: Splicing defect, I/NS: Indels out of frame/nonsense, GR: Genomic rearrangements KD: Kinase domain.

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
