# Peer review of "Two Novel FAM20C Variants in a Family with Raine Syndrome"

_genes, 2020, doi:10.3390/genes11020222_

Round 1
Reviewer 1 Report
This manuscript is a case report presenting 2 siblings with Raine syndrome. Different predicted pathogenic (frameshifting) heterozygous variants in FAM20C were identified in both parents, although DNA of adequate quality was not available to confirm transmission to either affected individual. The case report is supplemented by clinical photos, radiology, organ pathology and immunohistochemistry, the latter demonstrating convincingly that absense of immunostaining to FAM20C in one of the affected individuals, but not in a control. The authors also provide a comprehensive review of the literature on Raine syndrome.
Although novelty is relatively limited (the immunostaining result, report of missing Islets of Langerhans in the pancreas, and the fact that neither frameshifting variant has previously been reported in a case of Raine syndrome are more original features), this work represents a thorough report of a rare syndrome. I recommend that the authors pay attention to the following details:
Title: importantly, the authors did NOT demonstrate biallelic variants in the two siblings. Although this is likely to have been present, the evidence is indirect. Also, the word "variant" is generally preferred to "mutation" in this context, since no mutation was demonstrated to have taken place. Therefore I recommend that the title is modified to reflect this, for example to "Likely biallelic [or compound heterozygous] variants in two siblings with Raine syndrome".
General: (1) The remainder of the article should also be more careful in its wording regarding the two assumptions mentioned above.
(2) When referring to the gene, FAM20C should be italicised throughout (upright text when referring to the protein)
(3) For completeness the authors could refer to the recent case report on non-lethal Raine syndrome by Mamedova et al Calcif Tissue Int 105:567-72
line 170: and instead of y
line 179: The Genbank reference for the cDNA numbering used (NM_020223.3) should be given at the start of this section
line 182 Gly instead of Gli
Table 1: The authors have summarised all previously reported cases of molecularly analysed Raine syndrome. However they have copied the cDNA and amino acid numbering across from the original reports and some of these numberings are incorrect. Specifically I noticed that the apparently identical nucleotide substitution in the Simpson and Whyte papers appeared to corresponding to different amino acid substitutions. On further investigation, it appears that the Simpson numbering is incorrect - for example the substitutions at "Gly365" should actually be at Gly379. To avoid further confusion, when revising this table please can the authors check that all cDNA and amino acid numbering corresponds to the current Genbank reference FAM20C sequence.
lines 244-7: the authors have mixed up the maternal and paternal variants here. It is the paternal variant that was previously annotated in gnomAD, as described in Results
line 334: compound instead of compounds. This is an example of a place in the manuscript where the authors should state that the compound heterozygous state is "likely" or "predicted" (see comment on the title).
Author Response
Review 1
Title: importantly, the authors did NOT demonstrate biallelic variants in the two siblings. Although this is likely to have been present, the evidence is indirect. Also, the word "variant" is generally preferred to "mutation" in this context, since no mutation was demonstrated to have taken place. Therefore I recommend that the title is modified to reflect this, for example to "Likely biallelic [or compound heterozygous] variants in two siblings with Raine syndrome".
Response: That is correct, the biallelic variants in the two siblings could not be demonstrated. However, it is inferred that due to the variants identified in the parents correlate with the clinical data and pathological findings in the patients. We changed the title to: “An apparently biallelic combination of novel FAM20C variants in two siblings with Raine syndrome”
General:
The remainder of the article should also be more careful in its wording regarding the two assumptions mentioned above.Response: They was attended through the manuscript.
(2) When referring to the gene, FAM20C should be italicised throughout (upright text when referring to the protein)
Response: It was corrected through the manuscript.
(3) For completeness the authors could refer to the recent case report on non-lethal Raine syndrome by Mamedova et al Calcif Tissue Int 105:567-72
Response: The reference was considered.
line 170: and instead of y
Response: It was corrected (line 379).
line 179: The Genbank reference for the cDNA numbering used (NM_020223.3) should be given at the start of this section
Response: It has been integrated to the text.
line 182 Gly instead of Gli
Response: It was corrected (line 398).
Table 1: The authors have summarised all previously reported cases of molecularly analysed Raine syndrome. However they have copied the cDNA and amino acid numbering across from the original reports and some of these numberings are incorrect. Specifically I noticed that the apparently identical nucleotide substitution in the Simpson and Whyte papers appeared to corresponding to different amino acid substitutions. On further investigation, it appears that the Simpson numbering is incorrect - for example the substitutions at "Gly365" should actually be at Gly379. To avoid further confusion, when revising this table please can the authors check that all cDNA and amino acid numbering corresponds to the current Genbank reference FAM20C sequence.
Response: We reviewed all the variants, according to the number of Sequence ID: NM_020223.3. Indeed, Simpson numbering is incorrect, as well as Kochar. The table was corrected (line 560).
lines 244-7: the authors have mixed up the maternal and paternal variants here. It is the paternal variant that was previously annotated in gnomAD, as described in Results
Response: It was corrected (line 606).
line 334: compound instead of compounds. This is an example of a place in the manuscript where the authors should state that the compound heterozygous state is "likely" or "predicted" (see comment on the title).
Response: It was corrected through the text.
ADD: The lines had changed due to modifications.
Reviewer 2 Report
Minor comments
The manuscript title “Raine’s syndrome” should be “Raine syndrome”. Pg2, line 57; DMP4 is a mouse homolog of FAM20C. Therefore, “also called DMP4” should be deleted. Pg2, line 77; “imperfect” should be “imperfecta”. Pg2, line 87; “hypertrophic chondrocytes, osteoid formation” are description of normal cell type/tissue formation. Therefore, this sentence did not make sense to the reviewer. Pg3, line 113; didn’t authors perform genetic analysis using not only parents’ samples but also case subjects’? If so, please correct the sentence as Analyses were done on parents and case subjects. How about 1st 9-year’s daughter’s sample? Pg3, line 116; “5µm thick” should be “5µm of thickness”. Pg3, line 116; why only kidney and liver tissues were used? Please explain. Pg3, line 118; “dilute” should be “diluted”. Pg3, lines 123-124; if there was no quantification of immunohistochemistry was presented, evaluation of immunoreactivity in a blind manner is useless. The reviewer has major critiques on immunohistochemistry, please see comments below (in major comments). Pg3, line 125; there is no description of a 31 GW newborn control. Was this human subject non-diseased? How/why the pregnancy was terminated? Please explain. Pg3, line 133; authors stated there was gingival hyperplasia, however, the image provided was not large enough to verify its presence. Pg4, Figure 1 legend, line 151; It is confusing the numbering of family members as authors use the same number in both first and second generations. Case 1 has “2” in the second family and case 2 has “3” in the second family, however, in line 151, authors described that “case 1 and 2, II.1 and II.2 individuals respectively. These figure and legend should be clarified. Pg7, line 219; “and” is missing between choanal stenosis and gingival hyperplasia. Pg7, line 224; “cranium base” should be “cranial base”. Pg8, line 243; as authors did not show protein expression data (by, for example, Western blot analysis), “---that produces 206 and 185 amino acid proteins” was not accurate (this is assumption). Pg9, lines 247-250; the reviewer did not understand the logic in this sentence; 1) how did authors know this family’s case is monogenic entity?, 2) the reviewer did ot understand the sentence of “we couldn’t verify it due to DNA degradation of tissues from case 2 and unavailable tissue sample for case 1”. Please explain these points and re-write this sentence. Pg9, line 252; what did “SRN” indicate? Pg9, lines 271-272; “a broad spectrum of secretory pathway proteins (>100)”; what did authors mean by “(>100)”? Please re-write this sentence. Pg10, line 300; dentinogenesis imperfect phenotype was not detected in NLRS (actually it is difficult to determine this phenotype as dentin is covered by enamel unless tooth histology is performed). Please re-write this sentence. Pg10, lines 299-330; these three paragraphs were irrelevant to the current study as there was no investigation of teeth and/or other molecules listed in these paragraphs. The reviewer strongly suggest re-organizing the discussion section by removing these description.
Major comments
In this manuscript, authors stated repeatedly FAM20C protein expression (for example, Pg2-line 92, Pg3-lines 124-125, Pg6-line 200, etc.), however scientific method by this statement was based on immunohistochemistry. As immunohistochemistry is a non-quantitative method and only indirect immunoreactivity can be described, authors should not use the term of “protein expression”. In addition, as the specificity of the antibody used was not presented in this study, the reviewer could not determine whether the immunohistochemistry data shown in Figure 5 really represented FAM20C immunoreactivity. Although authors showed negative control data using PBS, this is not enough, rather authors should use species-matched (in this case rabbit) non-immune IgG as a primary antibody and perform the immunohistochemistry. When authors do this experiment, please note that the concentration of (NOT dilution) of anti-FAM20C antibody and non-immune IgG should be the same.
Alternatively if authors would like to state the protein “expression”, Western blotting should be performed as it is a gold standard of quantitative analysis.
In the same analogy, Pg10, lines 335-336, authors mentioned “an absent protein in liver and kidney” is not scientifically accurate statement. Immunohistochemistry can not describe the absence of protein as it is non-quantitative method (one can argue if authors use 1:10 of dilution, FAM20C immunoreactivity might be detected). Please avoid using the term of “absence” or presence of protein.
Author Response
REVIEWER 2
Comments and Suggestions for Authors
Minor comments
The manuscript title “Raine’s syndrome” should be “Raine syndrome”.
Response: It was modified to: An apparently biallelic combination of novel FAM20C variants in two siblings with Raine syndrome
Pg2, line 57; DMP4 is a mouse homolog of FAM20C. Therefore, “also called DMP4” should be deleted.
Response: It was deleted (line 106)
Pg2, line 77; “imperfect” should be “imperfecta”.
Response: It was corrected (line 125).
Pg2, line 87; “hypertrophic chondrocytes, osteoid formation” are description of normal cell type/tissue formation. Therefore, this sentence did not make sense to the reviewer.
Response: It was modified corrected (line 134-136).
Pg3, line 113; didn’t authors perform genetic analysis using not only parents’ samples but also case subjects’? If so, please correct the sentence as Analyses were done on parents and case subjects. How about 1st 9-year’s daughter’s sample?
Response: A sequence analysis of FAM20C exons was carried out in both parents and in paraffin embedded tissues from case 2. However, we didn’t obtain good quality electropherograms. Also, it was not possible to obtain 9-year’s daughter’s sample.
Pg3, line 116; “5µm thick” should be “5µm of thickness”.
Response: It was corrected (line 263)
Pg3, line 116; why only kidney and liver tissues were used? Please explain.
Response:
We used liver and kidney sections because many serum and plasma phosphoproteins (which are FAM20C targets), are synthesized and secreted from hepatocytes. Also, because liver is one of the tissues with higher expression (at least in adults).We used kidney because FAM20C participation is well described in regards to regulating excretion of phosphate through binding to FGFRs/α-KL in kidney. Also, kidney is another tissue with high expression (at least in adults).
Because of this, we thought we could more easily detect FAM20C immunoreactivity in both these tissues.
Pg3, line 118; “dilute” should be “diluted”.
Response: It was corrected (line 264).
Pg3, lines 123-124; if there was no quantification of immunohistochemistry was presented, evaluation of immunoreactivity in a blind manner is useless. The reviewer has major critiques on immunohistochemistry, please see comments below (in major comments).
Response: It was changed to “detection” through the text.
Pg3, line 125; there is no description of a 31 GW newborn control. Was this human subject non-diseased?
Response: It was from necropsy tissues of a 31 GW newborn, without Raine syndrome (no other data were available). It was modified (lines 263).
How/why the pregnancy was terminated? Please explain.
Response: The pregnancy was terminated because of the parents’ decision. They decide it because the prenatal ultrasonograpy findigs reported were compatible with Raine syndrome.
Pg3, line 133; authors stated there was gingival hyperplasia, however, the image provided was not large enough to verify its presence.
Response: Gingival hyperplasia was described since the first contact with the patient, by the Clinical geneticist (M). Maybe it’s not so evident in the picture (Fig 3C), but it was present and we consider it is a Raine syndrome feature.
Pg4, Figure 1 legend, line 151; It is confusing the numbering of family members as authors use the same number in both first and second generations. Case 1 has “2” in the second family and case 2 has “3” in the second family, however, in line 151, authors described that “case 1 and 2, II.1 and II.2 individuals respectively. These figure and legend should be clarified.
Response: It was modified (line 350).
Pg7, line 219; “and” is missing between choanal stenosis and gingival hyperplasia.
Response: It was corrected (line 346)
Pg7, line 224; “cranium base” should be “cranial base”.
Response: It was corrected (line 483).
Pg8, line 243; as authors did not show protein expression data (by, for example, Western blot analysis), “---that produces 206 and 185 amino acid proteins” was not accurate (this is assumption).
Response: It was modified to predicted proteins (line 605 and other sections of the manuscript).
Pg9, lines 247-250; the reviewer did not understand the logic in this sentence; 1) how did authors know this family’s case is monogenic entity?, 2) the reviewer did ot understand the sentence of “we couldn’t verify it due to DNA degradation of tissues from case 2 and unavailable tissue sample for case 1”. Please explain these points and re-write this sentence.
Response: The paragraph was modified (line 609)
Pg9, line 252; what did “SRN” indicate?
Response: It was corrected.
Pg9, lines 271-272; “a broad spectrum of secretory pathway proteins (>100)”; what did authors mean by “(>100)”? Please re-write this sentence.
Response: It was modified and integrated to a previous paragraph (line 731).
Pg10, line 300; dentinogenesis imperfect phenotype was not detected in NLRS (actually it is difficult to determine this phenotype as dentin is covered by enamel unless tooth histology is performed). Please re-write this sentence.
Response:
We modified the text. We add the references number 30 and 31, that report about amelogenesis imperfecta and dentinogenesis defects in non-lethal Raine syndrome (line 754)
Acevedo AC, Poulter JA, Alves PG, de Lima CL, Castro LC, Yamaguti PM, et al. Variability of systemic and oro-dental phenotype in two families with non-lethal Raine syndrome with FAM20C mutations. BMC Med Genet. 2015 Feb 21;16:8. DOI: 10.1186/s12881-015-0154-5 Elalaoui SC, Al-Sheqaih N, Ratbi I, Urquhart JE, O’Sullivan J, Bhaskar S, et al. Non lethal Raine syndrome and differential diagnosis. Eur J Med Genet. 2016 Nov;59(11):577–83. DOI: 10.1016/j.ejmg.2016.09.018
Pg10, lines 299-330; these three paragraphs were irrelevant to the current study as there was no investigation of teeth and/or other molecules listed in these paragraphs. The reviewer strongly suggest re-organizing the discussion section by removing these description.
Response:
It has been modified. We agree partially because indeed we didn`t evaluate other tissues or FAM20C targets. However, we think it is important to point that new FAM20C targets have been recently described and that they may have participation in Raine syndrome pathogenesis (lines 753-765).
Major comments
In this manuscript, authors stated repeatedly FAM20C protein expression (for example, Pg2-line 92, Pg3-lines 124-125, Pg6-line 200, etc.), however scientific method by this statement was based on immunohistochemistry. As immunohistochemistry is a non-quantitative method and only indirect immunoreactivity can be described, authors should not use the term of “protein expression”.
Response: It has been corrected to “detectable” and “not detectable” trough the text.
In addition, as the specificity of the antibody used was not presented in this study, the reviewer could not determine whether the immunohistochemistry data shown in Figure 5 really represented FAM20C immunoreactivity.
Response: The antibody used is manufactured by the Abcam Company (Cambridge, UK, catalog number ab154740) and has been previously referenced (Cozza G, et al., FEBS J. 2017 Apr; 284 (8): 1246-1257. doi: 10.1111 / febs. 14052 and Knab VM, et al., Endocrinology. 2017 May 1; 158 (5): 1130-1139. doi: 10.1210 / en.2016-1451).
Although authors showed negative control data using PBS, this is not enough, rather authors should use species-matched (in this case rabbit) non-immune IgG as a primary antibody and perform the immunohistochemistry. When authors do this experiment, please note that the concentration of (NOT dilution) of anti-FAM20C antibody and non-immune IgG should be the same. Alternatively if authors would like to state the protein “expression”, Western blotting should be performed as it is a gold standard of quantitative analysis.
Response: Negative immunostaining controls were initially incubated with PBS, instead of primary antibody, followed by secondary antibodies. Although we didn’t use species-matched non-immune IgG as a primary antibody (as suggested by you), which would have been the ideal control, the clarity of the staining patterns makes it very unlikely that the staining patterns observed are an artifact or spurious. We understand your comment and agree with it, but now we have no choice. Reviewer emphasizes that immunohistochemistry is not quantitative and mentions western blot as the gold-standard for quantitative protein immune assays. We disagree, as both assays are semiquantitative. For quantitative assays, the choice is ELISA, when possible. Moreover, we did not attempt to do quantitative immunodetection.
In the same analogy, Pg10, lines 335-336, authors mentioned “an absent protein in liver and kidney” is not scientifically accurate statement. Immunohistochemistry can not describe the absence of protein as it is non-quantitative method (one can argue if authors use 1:10 of dilution, FAM20C immunoreactivity might be detected). Please avoid using the term of “absence” or presence of protein.
Response: It was corrected id different line trough the text. It was changed to “dectable” o “not detectable”
ADD: The lines had changed due to modifications.
Round 2
Reviewer 2 Report
There are, unfortunately, several comments remained unanswered. Due to authors’ responses, some comments raised new critiques. There are comments from (a) to (k) below. Please revise carefully as this reviewer will not accept if these critiques are unanswered and will likely reject the manuscript.
Pg3, line 113; didn’t authors perform genetic analysis using not only parents’ samples but also case subjects’? If so, please correct the sentence as Analyses were done on parents and case subjects. How about 1st9-year’s daughter’s sample?Response: A sequence analysis of FAM20C exons was carried out in both parents and in paraffin embedded tissues from case 2. However, we didn’t obtain good quality electropherograms. Also, it was not possible to obtain 9-year’s daughter’s sample.
>If this reviewer understood well with your response, there was no genetic analysis for case 1, but only clinical diagnosis was made. In that case, the manuscript title “biallelic FAM20C mutation in two siblings” is not accurate. Please reconsider the title. Authors are also strongly advised to describe which human subject samples were analyzed more clearly under 2-2. Genetic analysis section to help readers better understand your work. Also, if there was no genetic analysis performed using case 1 sample, your description “case 1 was compound heterozygous” is overstating. Please re-write throughout the manuscript.
(b) Pg3, line 116; why only kidney and liver tissues were used? Please explain.
Response:
We used liver and kidney sections because many serum and plasma phosphoproteins (which are FAM20C targets), are synthesized and secreted from hepatocytes. Also, because liver is one of the tissues with higher expression (at least in adults).We used kidney because FAM20C participation is well described in regards to regulating excretion of phosphate through binding to FGFRs/α-KL in kidney. Also, kidney is another tissue with high expression (at least in adults).
Because of this, we thought we could more easily detect FAM20C immunoreactivity in both these tissues.
>Please describe your response in the manuscript.
(c) Pg3, line 125; there is no description of a 31 GW newborn control. Was this human subject non-diseased?
Response: It was from necropsy tissues of a 31 GW newborn, without Raine syndrome (no other data were available). It was modified (lines 263).
(d) How/why the pregnancy was terminated? Please explain.
Response: The pregnancy was terminated because of the parents’ decision. They decide it because the prenatal ultrasonograpy findigs reported were compatible with Raine syndrome.
>The comments (c) and (d) were about a 31 GW new born control sample. Please respond correctly. If this newborn was not diseased, please describe in the manuscript including the following point; your response answered this was not Raine syndrome newborn. How did you know?
(e) Pg3, line 133; authors stated there was gingival hyperplasia, however, the image provided was not large enough to verify its presence.
Response: Gingival hyperplasia was described since the first contact with the patient, by the Clinical geneticist (M). Maybe it’s not so evident in the picture (Fig 3C), but it was present and we consider it is a Raine syndrome feature.
>Reviewers solely judge the presence of clinical features by the data presented in the manuscript. As the data were not sufficient and not convincing, more convincing data were required for the initial review. Therefore, authors should have provided more convincing data/images. Please DO NOT refer a personal communication by your geneticist, which is not acceptable. If you cannot provide such convincing data, gingival hyperplasia phenotype should be removed.
(f) Pg9, line 252; what did “SRN” indicate?
Response: It was corrected.
>This is not corrected.
(g) New comment: pg10, line 416; your statement “total absence cause lethal phenotype” is overstating as this is impossible to prove and your data did not support such a cause-effect relationship.
(h) New comment: pg10, lines 425-428; BMP4 is not a member of SIBLING proteins. Please re-write.
(i) Pg10, lines 299-330; these three paragraphs were irrelevant to the current study as there was no investigation of teeth and/or other molecules listed in these paragraphs. The reviewer strongly suggest re-organizing the discussion section by removing these description.
Response:
It has been modified. We agree partially because indeed we didn`t evaluate other tissues or FAM20C targets. However, we think it is important to point that new FAM20C targets have been recently described and that they may have participation in Raine syndrome pathogenesis (lines 753-765).
>The review disagree with your response. Discussion part is not another part of introduction. It should discuss with your data. Therefore, these paragraphs should have been removed, rather you should discuss with any relevant information regarding brain, kidney, pancreas and liver, which you had data. This reviewer strongly advises in re-writing your discussion (or you should provide more thorough tooth and bone data, then keep the current discussion). Your response that “we think it is important to point that new FAM20C targets have been recently described and that they may have participation in Raine syndrome pathogenesis (lines 753-765).” can be justified if this manuscript is review article.
(j) In addition, as the specificity of the antibody used was not presented in this study, the reviewer could not determine whether the immunohistochemistry data shown in Figure 5 really represented FAM20C immunoreactivity.
Response: The antibody used is manufactured by the Abcam Company (Cambridge, UK, catalog number ab154740) and has been previously referenced (Cozza G, et al., FEBS J. 2017 Apr; 284 (8): 1246-1257. doi: 10.1111 / febs. 14052 and Knab VM, et al., Endocrinology. 2017 May 1; 158 (5): 1130-1139. doi: 10.1210 / en.2016-1451).
>According to the paper listed by your response, Cozza G et al. did not show Western blotting of FAM20C by anti-FAM20C antibody from Abcam. The data showed only anti-V5 tagged Western blotting, which did not support your justification that this Abcam antibody is specific to FAM20C.
Another paper by Knab VM et al. showed Western blotting data, however, their blot showed only experimental samples and no controls (no negative, positive controls in the blot). In addition, the FAM20C shown in Knab VM’s study was detected at around ~60kDa, whereas the original FAM20C paper published by Tagliabracci V et al. showed ~75kDa protein (Science, Science. 2012 Jun 1;336(6085):1150-3. doi: 10.1126/science.1217817.). Therefore, it is impossible to determine this antibody is specific to FAM20C. It would be, thus, important for your work to show the specificity by providing at least a proper negative control.
(k) Although authors showed negative control data using PBS, this is not enough, rather authors should use species-matched (in this case rabbit) non-immune IgG as a primary antibody and perform the immunohistochemistry. When authors do this experiment, please note that the concentration of (NOT dilution) of anti-FAM20C antibody and non-immune IgG should be the same. Alternatively if authors would like to state the protein “expression”, Western blotting should be performed as it is a gold standard of quantitative analysis.
Response: Negative immunostaining controls were initially incubated with PBS, instead of primary antibody, followed by secondary antibodies. Although we didn’t use species-matched non-immune IgG as a primary antibody (as suggested by you), which would have been the ideal control, the clarity of the staining patterns makes it very unlikely that the staining patterns observed are an artifact or spurious. We understand your comment and agree with it, but now we have no choice. Reviewer emphasizes that immunohistochemistry is not quantitative and mentions western blot as the gold-standard for quantitative protein immune assays. We disagree, as both assays are semiquantitative. For quantitative assays, the choice is ELISA, when possible. Moreover, we did not attempt to do quantitative immunodetection.
>As the authors admitted the importance of “ideal control”, please perform an additional experiment using species-matched non-immune IgG as a primary control (with the same concentration as anti-FAM20C antibody used in your experiment) and provide such data. This would very likely makes your data scientifically convincing and important. (For your information, please do not argue with what method is quantitative. This did not answer the question in the critique.)
Author Response
RESPONSE TO REVIEWER 2
Comments and Suggestions for Authors
There are, unfortunately, several comments remained unanswered. Due to authors’ responses, some comments raised new critiques.
There are comments from (a) to (k) below.
Please revise carefully as this reviewer will not accept if these critiques are unanswered and will likely reject the manuscript.
Pg3, line 113; didn’t authors perform genetic analysis using not only parents’ samples but also case subjects’? If so, please correct the sentence as Analyses were done on parents and case subjects. How about 1 9-year’s daughter’s sample?
Response: A sequence analysis of FAM20C exons was carried out in both parents and in paraffin embedded tissues from case 2. However, we didn’t obtain good quality electropherograms. Also, it was not possible to obtain the 9-year old daughters sample.
>If this reviewer understood well with your response, there was no genetic analysis for case 1, but only clinical diagnosis was made. In that case, the manuscript title “biallelic FAM20C mutation in two siblings” is not accurate.
Please reconsider the title.
Authors are also strongly advised to describe which human subject samples were analyzed more clearly under 2.2. Genetic analysis section to help readers better understand your work. Also, if there was no genetic analysis performed using case 1 sample, your description “case 1 was compound heterozygous” is overstating. Please re-write throughout the manuscript.
Response (a):
Title was reconsidered, it was changed to: Two novel FAM20C variants in a family with Raine syndrome. We couldn’t find the refered text: “case 1 was compound heterozygous” throughout the manuscript. We found the term “compound heterozygous” on three sections:
Lines 195:
“The two cases would have inherited these variants and therefore they appeared to be compound heterozygous.”
Line 263-269:
“We could not verify the presence of these variants in both cases due to DNA degradation of tissues in case 2 and unavailable tissue samples for case 1 as well as from the 9-year old daughter. Nevertheless, as RS has an autosomic recessive inheritance, we can assume that both variants were inherited together in cases 1 and 2 (compound heterozygous). If this was the case, these two variants represent a novel combination of biallelic FAM20C mutation in Raine osteosclerotic dysplasia and, therefore, they are new variants associated to lethal RS, although one of them had been previously reported in one heterozygous healthy individual.”
Lines 321-323
“These variants were apparently inherited as a biallelic combination in compound heterozygous, both of them causing a frameshift out of frame producing truncated proteins (185 and 207 aa).”
We added to section 2.2 (line 112), that genetic analyses were done in blood DNA from both parents. Also, we added that we didn’t have a sample from the 9-year old daughter in discussion section (lines 256-257). Besides, as variants were not detected in case 1 and 2, they were deleted in the table.
(b) Pg3, line 116; why only kidney and liver tissues were used? Please explain.
Response: We used liver and kidney sections because many serum and plasma phosphoproteins (which are FAM20C targets), are synthesized and secreted from hepatocytes. Also, because liver is one of the tissues with higher expression (at least in adults). We used kidney because FAM20C participation is well described in regards to regulating excretion of phosphate through binding to FGFRs/α-KL in kidney. Also, kidney is another tissue with high expression (at least in adults).
Because of this, we thought we could more easily detect FAM20C immunoreactivity in both these tissues.
>Please describe your response in the manuscript.
Response (b):
It was included. Lines 119-122.
(c) Pg3, line 125; there is no description of a 31 GW newborn control. Was this human subject non-diseased?
Response: It was from necropsy tissues of a 31 GW newborn, without Raine syndrome (no other data were
available). It was modified (lines 263).
(d ) How/why the pregnancy was terminated? Please explain.
Response: The pregnancy was terminated because of the parents’ decision. They decided it because the prenatal ultrasonography findings reported were compatible with Raine syndrome.
>The comments (c) and (d) were about a 31 GW new born control sample. Please respond correctly. If this newborn was not diseased, please describe in the manuscript including the following point; your response answered this was not Raine syndrome newborn. How did you know?
Response (c y d):
It was added the 31 GW newborn control, was deceased because of a non-syndromic congenital heart malformation. Line 118.
We know the 31 GW newborn was not affected by Raine syndrome because we counted with the clinical and pathological diagnosis.
(e) Pg3, line 133; authors stated there was gingival hyperplasia, however, the image provided was not large
enough to verify its presence.
Response: Gingival hyperplasia was described since the first contact with the patient, by the Clinical geneticist (M). Maybe it’s not so evident in the picture (Fig 3C), but it was present and we consider it is a Raine syndrome feature.
>Reviewers solely judge the presence of clinical features by the data presented in the manuscript. As the data were not sufficient and not convincing, more convincing data were required for the initial review. Therefore, authors should have provided more convincing data/images.
Please DO NOT refer a personal communication by your geneticist, which is not acceptable. If you cannot provide such convincing data, gingival hyperplasia phenotype should be removed.
Response (e):
It was deleted from lines 140, 153, 230.
(f) Pg9, line 252; what did “SRN” indicate?
Response: It was corrected.
>This is not corrected.
Response:
(f) It was corrected to RS.
(g) New comment: pg10, line 416; your statement “total absence cause lethal phenotype” is overstating as this is impossible to prove and your data did not support such a cause-effect relationship.
(h) New comment: pg10, lines 425-428; BMP4 is not a member of SIBLING proteins. Please re-write.
(i) Pg10, lines 299-330; these three paragraphs were irrelevant to the current study as there was no
investigation of teeth and/or other molecules listed in these paragraphs. The reviewer strongly suggest reorganizing the discussion section by removing these description.
Response: It has been modified. We agree partially because indeed we didn`t evaluate other tissues or FAM20C targets. However, we think it is important to point that new FAM20C targets have been recently described and that they may have participation in Raine syndrome pathogenesis (lines 753-765).
>The review disagree with your response. Discussion part is not another part of introduction. It should discuss with your data. Therefore, these paragraphs should have been removed, rather you should discuss with any relevant information regarding brain, kidney, pancreas and liver, which you had data. This reviewer strongly advises in rewriting your discussion (or you should provide more thorough tooth and bone data, then keep the current discussion). Your response that “we think it is important to point that new FAM20C targets have been recently described and that they may have participation in Raine syndrome pathogenesis (lines 753-765).” can be justified if this manuscript is review article.
Response (g-i):
(g) It was modified (“total absence cause lethal phenotype”), to “Therefore, the total absence of any functional form of this protein could cause the lethal phenotype.” Line 278-279.
(h) It was re-written. Line 289-292.
(i) We deleted the refereed discussion part.
(j) In addition, as the specificity of the antibody used was not presented in this study, the reviewer could not determine whether the immunohistochemistry data shown in Figure 5 really represented FAM20C immunoreactivity.
Response: The antibody used is manufactured by the Abcam Company (Cambridge, UK, catalog number
ab154740) and has been previously referenced (Cozza G, et al., FEBS J. 2017 Apr; 284 (8): 1246-1257. doi: 10.1111/ febs. 14052 and Knab VM, et al., Endocrinology. 2017 May 1; 158 (5): 1130-1139. doi: 10.1210 / en.2016-1451).
>According to the paper listed by your response, Cozza G et al. did not show Western blotting of FAM20C by anti- FAM20C antibody from Abcam. The data showed only anti-V5 tagged Western blotting, which did not support your justification that this Abcam antibody is specific to FAM20C.
Another paper by Knab VM et al. showed Western blotting data, however, their blot showed only experimental
samples and no controls (no negative, positive controls in the blot). In addition, the FAM20C shown in Knab VM’study was detected at around ~60kDa, whereas the original FAM20C paper published by Tagliabracci V et al. showed ~75kDa protein (Science, Science. 2012 Jun 1;336(6085):1150-3. doi: 10.1126/science.1217817.).
Therefore, it is impossible to determine this antibody is specific to FAM20C. It would be, thus, important for your work to show the specificity by providing at least a proper negative control.
Response (j):
The paper by Cozza, indeed has no negative control for western blot, but describes in reagents section the Abcam anti-FAM20C.
The paper by Knab VM showed Western blot (no negative control) and IHC (but only describes that negative control was performed by omitting primary antibody).
We regret we don’t have a proper negative control. As FAM20C has ubiquitous expression, we have no tissues with no FAM20C expression, like tissue from a FAM20C knockout animal or other sample types.
(k) Although authors showed negative control data using PBS, this is not enough, rather authors should use
species-matched (in this case rabbit) non-immune IgG as a primary antibody and perform the immunohistochemistry.
When authors do this experiment, please note that the concentration of (NOT dilution) of anti-FAM20C antibody and non-immune IgG should be the same. Alternatively if authors would like to state the protein “expression”, Western blotting should be performed as it is a gold standard of quantitative analysis.
Response: Negative immunostaining controls were initially incubated with PBS, instead of primary antibody, followed by secondary antibodies. Although we didn’t use speciesmatched non-immune IgG as a primary antibody (as suggested by you), which would have been the ideal control, the clarity of the staining patterns makes it very unlikely that the staining patterns observed are an artifact or spurious. We understand your comment and agree with it, but now we have no choice. Reviewer emphasizes that
immunohistochemistry is not quantitative and mentions western blot as the gold-standard for quantitative protein immune assays. We disagree, as both assays are semiquantitative. For quantitative assays, the choice is
ELISA, when possible. Moreover, we did not attempt to do quantitative immunodetection.
>As the authors admitted the importance of “ideal control”, please perform an additional experiment using speciesmatched non-immune IgG as a primary control (with the same concentration as anti-FAM20C antibody used in your experiment) and provide such data. This would very likely makes your data scientifically convincing and important. (For your information, please do not argue with what method is quantitative. This did not answer the question in the critique.)
Response
(k) We regret it is not feasible to do additional IHC negative controls, using species-matched non-immune IgG as a primary control, because of lack of sufficient material from tissue blocks.
Round 3
Reviewer 2 Report
There are mainly two issues which remained unsolved in this manuscript/work. The following are the reasons leading to reject this manuscript in the current form.
FAM20C mutations were not confirmed in case 1. Ideally, 9-year old child’s genetic analysis should have been performed as well. Genetic analysis is thus incomplete, making this work preliminary. There is no “ideal” negative control for immunohistochemistry provided by authors, although there were two opportunities to revise. As authors did not provide such a negative control, it is difficult to make any conclusion.I regret having to send you a negative decision, as I can appreciate the time and effort you have expended in carrying out your study.